# A Comparative Assessment of JVM Frameworks to Develop Microservices

Łukasz Wyciślik [1,*], Łukasz Latusik [2] and Anna Małgorzata Kamińska [3]

1    Department of Applied Informatics, Faculty of Automatic Control, Electronics and Computer Sciences, Silesian University of Technology, 44-100 Gliwice, Poland
2    Independent Researcher, 43-100 Tychy, Poland
3    Institute of Culture Studies, University of Silesia in Katowice, ul. Uniwersytecka 4, 40-007 Katowice, Poland
*    Correspondence: lukasz.wycislik@polsl.pl

**Abstract:** With the ever-increasing wide spread of the Internet, the number of web services, web applications, and IoT devices is growing every year. This brings a number of challenges, both in terms of network bandwidth and the ability to scale individual computing nodes, whether they are large systems running in computing clouds or smaller IoT devices running closer to their data sources (so-called edge computing). In both cases, the way to cope with handling large numbers of users/requests is horizontal scaling, the implementation of which today is using the concept of microservices. However, the concept itself is not enough—we need ready-made application frameworks that allow us to easily implement and deploy efficient services. In the case of the Java ecosystem, which is one of the most mature platforms for enterprise-class software development, several frameworks dedicated to the development of microservices have been engineered recently. These tools support system developers in implementing communication, computation, and data storage mechanisms. However, so far, there is a lack of comparative analysis of individual solutions in the scholarly discourse to assess their performance and production maturity, so the authors in this article try to fill this gap. Based on synthetic tests developed by the authors, the most promising frameworks (Spring Boot, Micronaut, Quarkus) were analyzed both in terms of computational, compilation, or deployment performance. The results obtained can help system architects make rational and evidence-driven choices of system architecture and technology stacks.

**Keywords:** microservices; scaling; JVM; spring boot; quarkus; micronaut; benchmarks



## 1. Introduction

With the increasing prevalence of Internet availability in digital societies and the ongoing Industrial Revolutions 4.0 and 5.0, the demand for computing power is growing every year [1]. Providers of global commercial Internet services (such as Netflix, LinkedIn) or IoT related platfroms (such as Bosch IoT Suite, Samsung SmartThings Cloud) but also government agencies providing information services to their citizens countrywide (for example, e-Health) and corporations (for example, e-Banking) face a particular challenge when it comes to ensuring the continued availability of digital services they provide with ever-increasing user volumes [2]. These issues are related to the so-called scaling of computer systems, understood as guaranteeing a constant level of quantitative quality indicators of the use of a given system with an increasing number of its users or requests to it. More requests per unit of time mean, of course, a greater demand for computing power, which is most easily provided by adding more CPU cores or RAM to the computing node. This is known as vertical scaling, and while it seems the simplest, it is limited by the architecture of computers being built today. Its opposite is horizontal scaling, which involves increasing the number of computing nodes of a given computer system [3]. This is, of course, much more complicated, as it requires additional overhead for orchestrating

the entire system—i.e., provisioning computing nodes, deploying successive instances of services on them, monitoring the load, and, finally, a specific approach to building software that gives little overhead for deployment and running. This is particularly important for two reasons. The first is to support scaling server-based systems (which provide their services to other clients including IoT devices) deployed in large data centers. The second is to increase the computing power of systems run at the edge (so-called edge computing), where individual computing nodes, often supporting IoT devices or being IoT devices themselves, have limited capacity.

The emergence of an architectural style of software development and deployment called microservices [4] can be considered a milestone in the developing of horizontal scaling techniques for computer systems. The main principles behind the microservices approach are to divide a complex system into smaller self-containing, lightweight, independently deployable executable units that can be easily instantiated in multiple copies in areas of increased throughput demand when needed. A big significant advantage in deploying microservices is the ability to containerize them, that is, to package executable code with all dependencies and a minimalist operating system into a single file (a so-called image), which can then be deployed and run in multiple instances on the host operating system and even in a cluster. The most popular containerization standard at the moment is Docker, and the cluster used to host containers that allow them to scale dynamically is Kubernetes [5]. However, their use is not a prerequisite for building a system that follows the microservices approach.

The need to scale services along with the ability to scale them dynamically (autoscaling) [3] when system load variability is challenging to predict makes the cloud computing model particularly interesting. This is because the owner of the system does not have to worry about the infrastructure, and does not need to buy it in advance, but can use the infrastructure provided by a third party and pay for it in a pay-as-you-go model, i.e., only for the computing resources actually allocated and consumed. The cost-effectiveness of such an approach will depend mainly on the software technologies used to build the computer system and the auto-scaling mechanisms employed. Depending on the variability of the system load, not only the overall performance of software components, but also the time to instantiate and run a single copy of a microservice, or the time to respond to a first-time request (cold start) may be critical here.

One of the most mature and versatile ecosystems for software development is Java. With its Spring Boot application framework, it was one of the first to start supporting the development of applications that follow a microservices approach. However, Java as a multipurpose environment carries features that are not always optimal for microservice systems. These features include, for example, interpreting source code, running programs on a virtual execution environment (JVM) intended to ensure the portability of programs between different computer architectures, etc. Recently, these problems have been recognized, resulting in the development of frameworks and solutions for the Java ecosystem strictly dedicated to develop microservices [6]. Thus, the goal of this work is to assess the most popular microservices development frameworks currently for the Java ecosystem, in order to enable the system architects to make design decisions based on rationales drawn from the outcomes of this study.

Admittedly, such topics are occasionally covered in industry blogs and publications, but in the scientific discourse, articles on the subject are few, and the research conducted is quite superficial and usually based on peculiar problems and applications. Therefore, the authors, as part of the research described in this paper, developed their own synthetic test framework to evaluate individual solutions in terms of the most common challenges encountered when implementing real microservices-based systems.

This paper is organized as follows: Section 2 gives the background of the research being conducted in terms of an introduction to cloud computing and microservice issues in general, as well as technical means supporting the development of the latter. Related studies are described there as well. Section 3 explains the applied methods, i.e., the composition

of the testbed, the selection of application frameworks under test, and the criteria of comparison, while Section 4 presents the results being obtained. Section 5 interprets the results obtained and presents the limitations of the research and further possible directions for it. The last section is the conclusion.

## 2. Background

As cloud computing has grown in popularity, competition among cloud providers has increased, resulting in, among other things, a wide range of ways to charge for computing power consumed/allocated. On the one hand, the lowest unit prices can be obtained by reserving computing power in advance for a longer period of time, but on the other hand, many currently deployed systems have a volume of users that is difficult to predict, and the high load of the system is only periodic, usually based on daily, weekly or monthly intervals. In such cases, the long-term allocation of computing resources to handle pile-ups over the system's lifetime is usually economically unjustifiable. Therefore, mechanisms are introduced into the system's architecture to enable autoscaling, and the effectiveness of that autoscaling depends on the software means used.

However, the world of programming technologies is vast and still expanding rapidly. The choice of technology is a significant architectural decision. The base one, which is the foundation of each project or system, is the programming language. This choice binds to the given tools available within the ecosystem/platform or a runtime environment.

None of the microservices, which are presently the basic concept behind horizontal autoscaling, is created directly from scratch—this would be inefficient, time-consuming, and error-prone. From this point of view, the essential part is a framework (or in some cases library) that will be a foundation for system development and is strictly connected to the chosen language platform. Depending on the ecosystem, frameworks have different names or represent different approaches. For example, for the C# language, it is ASP.NET, for the JVM platform it is Spring, while for Node.js it is Express.js. There are many, many others for subsequent ecosystems, but the main purpose of them remains the same–to allow leveraging one of the fundamental software design principles—DRY (Don't Repeat Yourself) [7] by providing designers and developers with an application structure and ready to use solutions for common use cases such as data access, security, communication, creation of API endpoints and many more. When choosing the technology stack and framework that will be used for microservices within the given system one should consider several aspects such as community support, popularity, usefulness but also performance.

With the proliferation of IoT-class devices, research began to be conducted, in which scientific interest has not waned to this day, on communication protocols to ensure the integrity and security of messages being transmitted, while at the same time ensuring the lightness of those messages which is required by the relatively low computing capabilities of IoT devices. Moreover, the occupation of considerable transmission bandwidth resulting from the multitude of IoT devices motivates efforts to develop protocols concise in communication. At the same time, some shortcomings of the communication protocol developed in the SOA (service oriented architecture) era for communication between individual network services (SOAP) were recognized, resulting in the RESTful approach, which is also eagerly being adopted in the IoT world today.

More and more models are being proposed for IoTs, where middleware is used to expose device data through REST API and to hide implementation details, and act as an interface for the user to interact with sensor data [8,9]. Research on the use of the REST approach in IoT sensors is carried out in many areas of everyday life supported by IT, such as healthcare [10–12], agriculture [13–15], smart cities [16–19], smart grids [20,21], and more. In addition to communicating with individual IoT devices or other network services, the middleware layer often aggregates and processes the collected data and stores it in databases. As these three areas are crucial for implementing IoT and web services interoperability, therefore the authors of this article will evaluate selected JVM frameworks focusing mainly on them.

In order to evaluate and compare the performance of given technologies, it is necessary to test them in a controlled runtime environment based on identical workloads generated by either a ready-made sample application or an application developed specifically for testing. This is the approach taken in this article, as well as in related articles to the research presented here. There have already been conducted a scientific comparative analysis of the Java ecosystem frameworks [22] (specifically Spring Boot and Play) around the above-mentioned three axes. However, the authors there did not focus on applications for building microservices, but for web apps and the results they obtained do not clearly indicate a winner. On the other hand, published research conducted in 2020 on new, but already well-established, solutions [23]—Quarkus and GraalVM—is largely descriptive, and its benchmarking part (using Renaissance Suit), focuses mainly on computing load, leaving out communication and database access aspects. The results obtained by the authors are promising for the GraalVM solution, and the authors propose to start using this platform in production environments. Empirical studies related to so-called green computing, that is, among other things, the energy efficiency of computing, are also being undertaken. In the case of the Java ecosystem, tests have been conducted for various benchmark applications with and without the use of the Spring framework [24]. The results clearly show the benefits of using the Spring framework.

It may also be interesting to ask whether a microservice architecture applied to solutions that do not require auto-scaling will bring any advantages over a traditional monolithic architecture. This was tested by the authors [25] who studied the Spring Boot framework in this application, and the results indicate that with expected low application load, it may not make sense to go for the microservices approach, as the gain, if any, may be small, while the complexity of building a microservices solution is always higher. In another study, the authors [26] ask themselves about the performance of technologies for synchronous and asynchronous communication between microservices. Their results suggest that under light load synchronous communication is sufficient, while when a fixed number of microservice instances can no longer handle the traffic it is worth reaching for queuing mechanisms. However, the question may arise—why not take advantage of the possibility of scaling by increasing the number of instances in such a case? In addition, the authors do not specify what technology was used to implement synchronous communication and whether it was based on a reactive or blocking approach.

Finally, the authors of a 2020 article titled "Implementation of the Internet of Things Application Based on Spring Boot Microservices and REST Architecture" [27] demonstrated their proof-of-concept solution and argued that the Spring Boot framework is particularly useful for implementing modern IoT solutions. Moreover, a recent paper by Plecinski et al. [28] touches on the performance topics of the application frameworks studied in this article (i.e., Spring Boot, Quarkus and Micronaut frameworks). However, the authors of that paper used the ready-made application as the subject of their performance research and did not look individually at the performance of the individual component mechanisms of typical microservices applications, such as the communication layer, the data layer or the computation layer.

Findings of the two works mentioned above, confirming the suitability of Java Spring Boot framework to implement microservices, together with the constant evolvement of direct competitors of Spring Boot (such as Quarkus or Micronaut) have generated the authors' motivation to undertake studies on this topic.

## 3. Materials and Methods

Within the available research we can see a growing interest and usage of the microservice-based platforms that serve as a backbone and foundation for applications using IoT devices. Notwithstanding, at the same time, a lack of standardization or some widely accepted application model that could serve as a reference for performing a comparative analysis of application frameworks, an executable form of application, runtimes, cloud environments or design patterns used in the microservice oriented systems can be noticed. Whereas

in the case of other technological solutions, such as databases, that standardization can already be found (e.g., TPC Benchmarks [29]) and even tools that use those standards (e.g., HammerDB).

However, in terms of such a research perspective, there is a common denominator for the application models used to analyse these factors. In most cases, application models simulated the equivalent of an entire computer system. This approach has the significant benefit of accurate representation of reality, which may positively impact the achieved results. Despite that, it has drawbacks as well. Maintenance of such applications and their deployments to the test environment is expensive and requires much time. It may also discourage further research development.

### 3.1. Adopted Model of Application

The mentioned earlier evident lack of an adopted model for the application that would be widely used in the research community for comparative analysis has shown an apparent need to provide some standard for the application model and tests being carried out. These factors and the need for evaluation of the frameworks in the context of this paper influenced the preparation of own model for the testing application (https://github.com/latusikl/jvm-frameworks-comparative-assessment, accessed on 30 December 2022).

The application model definition was based on the API specification (in OpenAPI 3.0 format) that should be implemented in the microservice created with the given framework. A graphical representation of the API description is presented in Figure 1. All of the 11 endpoints provided in the specification can be divided into 4 main categories.

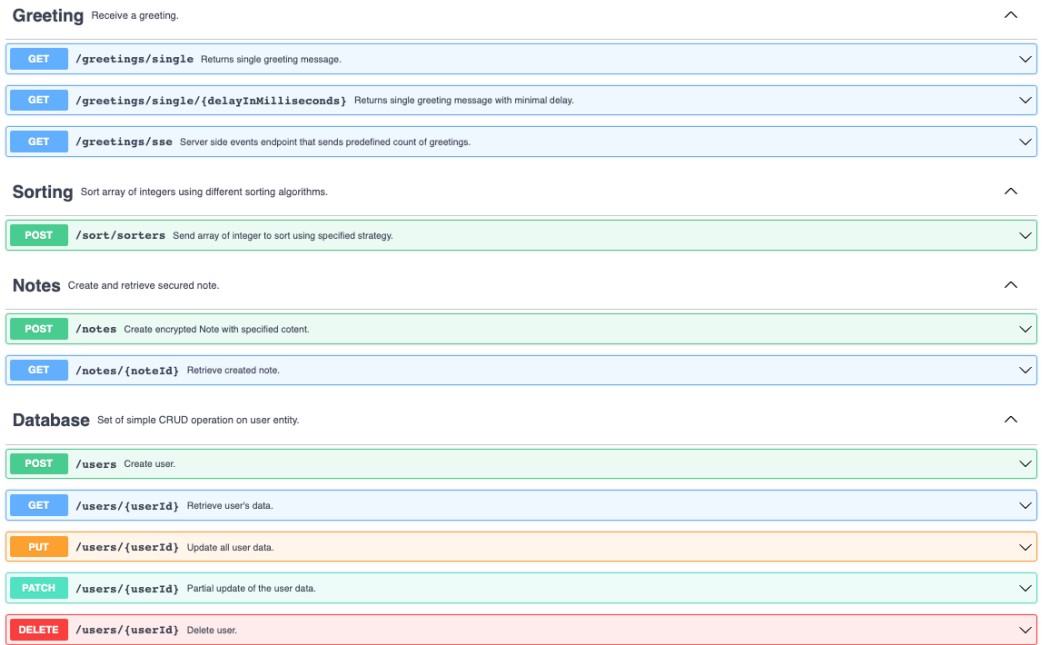

**Figure 1.** Graphical version of API description.

The first category is so-called "Greeting Operations". It is a set of simple HTTP GET requests and Server-Sent Events that return the greeting message. The purpose of this category is to check how the application behaves during the load generated by operations that do not demand a noticeable amount of computing resources and do not need external resources such as data from other microservices, database access, queuing systems utilisation, etc.

The second category is called "Sorting Operations". It is represented by a single endpoint that allows executing sorting of Integer arrays using one of three implemented sorting algorithms that have the same average time complexity:

- Quicksort,

- Merge Sort,
- Heap Sort.

The purpose of this operation category is to check how a given application behaves in case of a synchronous request that requires the execution of calculations that demands a noticeable amount of time and computational power.

The third category, "Notes Operation", is a set of two endpoints that allows the creation and retrieval of the encrypted note using the AES (advanced encryption standard) algorithm. After encryption, note content is stored as a file in external object storage, e.g., AWS S3 or Google Cloud Storage. A note is saved and retrieved by executing HTTP requests between tested microservice and external resources using the framework's HTTP client.

The goal for this category is to evaluate provided by the framework's HTTP clients that allow the execution of external calls. In the case of the system in the microservice architecture it plays a meaningful role and is commonly used to communicate with other parts of the system.

The last category, "Database Operations", is a set of CRUD (create, read, update and delete) operations that give the possibility to manage a simple entity that contains primary user data. This operation involves the usage of an external resource—a data store. A relational SQL database was used for tests in favour of the NoSQL one. SQL databases are still one of the most commonly used and most popular, and even they are reincarnated with the NewSQL trend [30].

The purpose of this category is to check how the frameworks handle object-relational mapping and database communication, as almost every microservice will probably involve some data persistence.

*3.2. Frameworks under Test*

The choice of the frameworks being tested was not left to chance. Based on the selection criteria described below, frameworks applicable to the Java ecosystem have been selected for the comparison.

The first obvious criterion was the possibility of microservices with REST API creation using the Java programming language, which can be called the most popular in the JVM environment.

The second criterion is the result of the yearly growing popularity of open-source software. This type of software is an element of almost every project or product. As a result of that trend, only open-source-based frameworks were considered.

The third one taken into consideration was the popularity of the given solution. It is an essential factor that benefits any project/product with the given framework as a microservice foundation. It directly impacts the size of the related community but also the quality of community support which is an indispensable help in terms of development, maintenance, and security fixes for the foundation of our project/product. As a measure of popularity, several factors were taken into account. Factor number one was the number of "stars" received by the project on the GitHub platform. This value directly relates to the project's popularity and can be seen as its denominator [31,32]. We can even see its usage in some research [33]. In the case of the project consisting of multiple repositories, the one with the most stars were considered. Factor number two was the popularity of the framework within the developer's community based on the survey's results carried out by two recognizable IT companies—Snyk.io and JetBrains [34–37]. The mentioned surveys carried out by these two companies were conducted on a significant number of respondents, which allows us to consider these surveys as showing trends within the Java ecosystem. Regarding Snyk.io, the 2020 and 2021 surveys were conducted on approximately 2000 Java developers. In the JetBrains reports, the number of respondents varied more significantly depending on the year the study was conducted. In 2020, it was around 20,000 respondents, and in 2021, around 31,000 respondents.

The last is the possibility of robust support for using the alternative executable form to traditional JAR file–GraalVM native images [38]. This criterion was dictated by the declared

benefits of the application in an alternative form, such as a significantly shorter start-up time and smaller size of the packed application, which in the world of microservices can have a very positive impact on the auto-scaling mechanism. Based on that criterion, the aspect of the impact of executable form was taken into consideration when executing tests and presenting results.

Applying the adopted criteria presented above, the first sure choice for the analysis was Spring Boot. Over 50% of the respondents in each survey declared they used it. It can also be named the most popular based on the GitHub stars criterion. It provides GraalVM support as well. When considering other frameworks to compare, such as Dropwizard, Struts, Vaadin, MicroProfile, Micronaut and Quarkus, only two offered robust support for GraalVM in combination with a similar application creation approach to the Spring Boot. Moreover, those two frameworks can be described as having a position as the direct rivals for Spring and having significant interest and community usage. Based on that, the following frameworks were chosen for comparison:

- Spring Boot (https://spring.io/projects/spring-boot, accessed on 30 December 2022),
- Micronaut (https://micronaut.io/, accessed on 30 December 2022),
- Quarkus (https://quarkus.io/, accessed on 30 December 2022).

All three frameworks support the reactive programming model. In the case of Micronaut and Quarkus it is a foundation, whereas in Spring Boot, it is available in the form of the Spring WebFlux. Applications in frameworks were implemented using reactive programming and libraries such as Project Reactor (https://projectreactor.io, accessed on 30 December 2022) or Mutiny (https://smallrye.io/smallrye-mutiny, accessed on 30 December 2022). It is also worth noticing that for the moment GraalVM support in Spring Framework is in the beta phase. The enablement of first-class support based on the beta support will be available with the upcoming Spring Framework 6.0/Spring Boot 3.0 release.

### 3.3. Comparative Criteria

It can be seen that most comparative analysis that concerns similar topics use a set of performance tests to evaluate and compare analysed solutions. This approach was followed in this case as well.

A foundation of the evaluation criteria for the problem at hand is a set of performance tests executed according to the defined scenarios and in controlled environment to provide the closest possible condition for each framework. In the case of the performed tests, the following metrics were taken into account:

- average response time,
- number of successful and failed executions of the test scenario,
- 0.5/0.75/0.95 quantile and standard deviation of response time,
- CPU and RAM usage,
- number of scenario executions per time unit.

Tests were implemented using the Gatling framework (https://gatling.io/, accessed on 30 December 2022). The main reasons behind the chosen tool were the usage of the JVM platform, ease of writing tests, the possibility of testing Server-Sent Events and clear report generation.

Taking into account the specifics of the issue being raised, additional factors were considered that matter in the microservice world. Besides performance tests, the following metrics were taken into account during the comparison:

- application startup time (measured by the tested framework itself),
- compilation time,
- size of the Docker container,
- size of the application executable file.

All these criteria have a direct or indirect impact on the microservices systems. Some of them, such as startup time or application size, can impact the autoscaling mechanism,

and its optimization can be beneficial. Others, such as response time and resource usage, directly affect the operational costs and the number of nodes needed during high load.

### 3.4. Test Scenarios

For performance tests, seven scenarios have been defined. They can be divided into two main categories—stress and average load tests. Each execution of tests uses a unique data set generated using randomness. A brief description of the scenarios is presented in Tables 1 and 2.

**Table 1.** Description of stress test scenarios.

| Scenario Name | Virtual Users | Short Description |
|---|---|---|
| SingleGreeting | 20,000 | Execute a single HTTP GET request for a single greeting and check the response. |
| GreetingSse | 15,000 | Open SSE connection, wait 10 s for events then close connection and check response. |
| CreateFetchDelete | 20,000 | Create a user and verify if the user was created. In the next step, fetch the user from the system and check if the data are correct. Then remove the user and verify if the user was removed by trying to fetch user data from the system. |
| MediumNumberSet | 5000 | Sort table using the merge sort algorithm then verify the result. Next sort another table with the same size using the Quicksort and verify the result. At the end choose another set of test data and sort table using heap sort and verify the result. The table that will be sorted has 10,000 elements. |

**Table 2.** Description of average load test scenarios.

| Scenario Name | Short Description |
|---|---|
| SmallNumberSet | The test scenario works the same as in the case of MediumNumberSet. The table size is reduced to 2000 elements. |
| Notes | Create a note with random content, then check if the key and the id of the created note were returned. Fetch created note using provided in the previous step key and id. In the end, verify it the received note text (after decryption) matches the uploaded one. |
| CreateFetchPatchDelete | The test scenario also operates on the user entity and is similar to the CreateFetchDelete. The difference is that it is extended with the user data modification and re-verification after that. The data update procedure is executed after fetching data of the created user and before user removal. |

Performed stress tests are designed to generate a high load on the application that causes high CPU usage within the upper limits of the virtual machine possibilities. Full load predicted during these scenarios appears immediately, and there is no delay between subsequent requests that would imitate the user's behaviour. These tests use a fixed number of virtual users that perform the specified actions in the loop during a specified test duration time. A number of virtual users were selected experimentally for each scenario, increasing the number of users by 2500 after each trial to match the load to the allocated resources and highlight possible differences between the frameworks while maintaining

the adopted criteria of high resource utilization. The time for each test was constant at 10 min which is a compromise between the duration of the experiment and the possibility of highlighting probable problems during more prolonged exposure to high load.

The purpose of the performed average load tests was to compare how applications behave under the average load. These tests have the same condition defined independently from the test scenario. Short breaks between requests are ensured with a random distribution of length within the specified range (from a few to several seconds) that imitates user behaviour or time between user actions. For this category, the number of test execution is constant in contrast to the number of virtual users that changes during the test according to the specified criteria—at the beginning, 250 concurrent users executing the scenario. Then 500 users spread out evenly over one minute, and lastly, a constant number of 30 users per second for the next 2 min. These conditions were experimentally selected and allowed us to achieve average utilization of allocated resources while maintaining differentiation in the traffic that was generated.

### 3.5. Test Environment

All performance tests were executed using the same cloud environment with the same resource allocation. Due to previous experience with the platform, as a cloud provider, Google Cloud was chosen to conduct the experiments. This choice was not crucial, however, as generic computing resources (VPS, referred to by GCP as Compute Engine) were used to conduct the experiments, and no services specific to the chosen cloud computing provider were used. It could be easily replaced with another provider. To provide isolation and constant resources application and test machines were launched using Linux-based virtual machines with allocated (non-shareable) resources inside VPC (virtual private cloud) to provide direct HTTP connection between machines—machine resource configurations are presented in Tables 3 and 4. External resources such as database and cloud storage were assigned as well. A simplified schema of the test bed is presented in Figure 2.

**Table 3.** Test executor virtual machine parameters.

| CPU | RAM | Disc Size |
|---|---|---|
| 2 × vCPU Intel Broadwell | 8 GB | 10 GB |

**Table 4.** Application virtual machine parameters (e2-standard-2).

| CPU | RAM | Disc Size |
|---|---|---|
| 6 × vCPU Intel Cascade Lake | 128 GB | 100 GB |

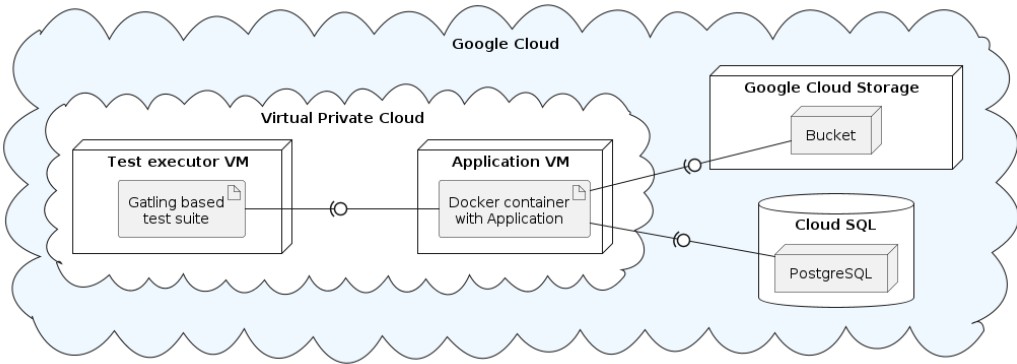

**Figure 2.** Deployment diagram of test bed.

### 4. Results

This section shows the results of the experiments that were carried out, starting from the general metrics that did not relate to the test scenarios that were executed and then

going into details about the results for each test scenario. For each of the frameworks, the results were measured for two possible executable forms—the standard and commonly used multiplatform JAR file and the native (platform-specific) executable that does not require a JVM environment to run produced by the usage of GraalVM. These results have been presented as separate charts or separate data series in the illustrated charts. For the JAR executable form they are marked with the "JAR" suffix and for the native executable they are marked with the suffix "native image" or "native".

### 4.1. Compilation Time

Five compile-time trials were performed. The results were averaged and presented in Figure 3 along with min and max values. Compilation time was taken based on the time reported by the Gradle build tool (used for all frameworks) as the execution time for the compilation task.

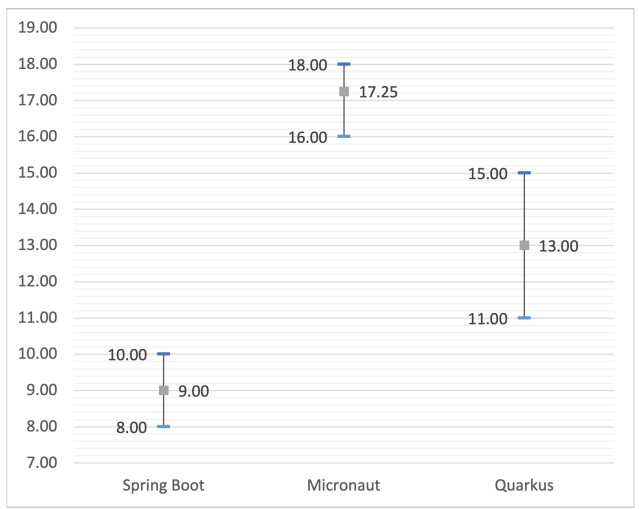

(**a**) JAR executable.

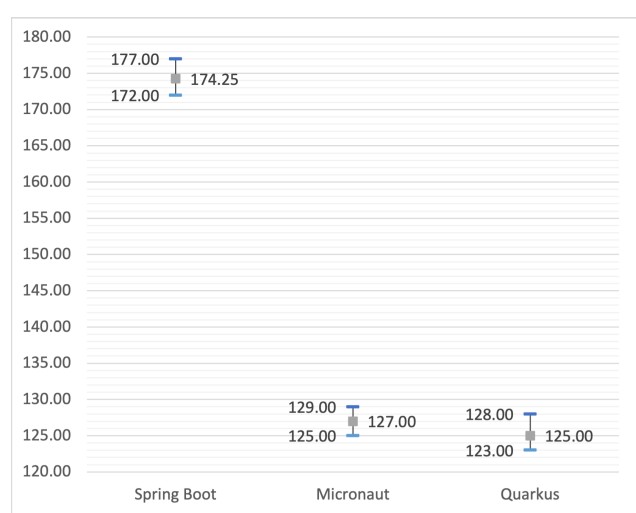

(**b**) Native image.

**Figure 3.** Average compilation time [s].

### 4.2. Startup Time

As in the case of the compilation time in the application's startup time, five trials were performed, and the results were averaged (Figure 4). The measurements were taken based on the startup time declared by each framework in logs.

### 4.3. Docker Image and Executable File Size

Docker images were built using the same base image for each executable form using a multistage build. For the final image, the Debian bases images were used as the foundation (openjdk:17-slim-bullseye or debian:bullseye). In terms of JAR files, it is worth mentioning that for the Quarkus, the recommended "fast-jar" approach was used. The whole directory generated with executable was considered, as the entire content in this directory is needed to run the application. Achieved results are presented in Figure 5.

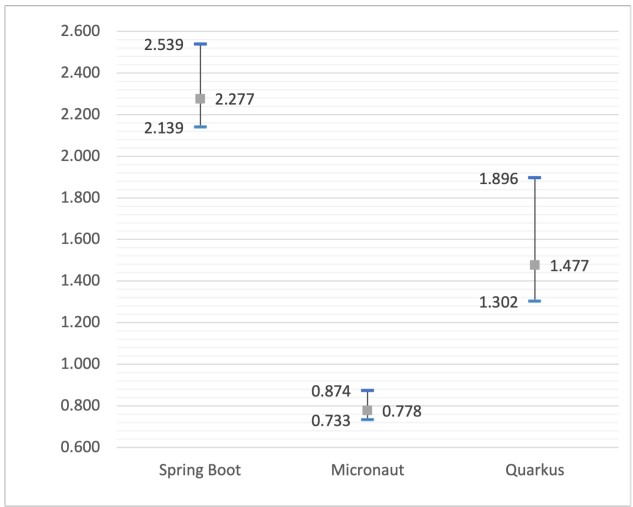

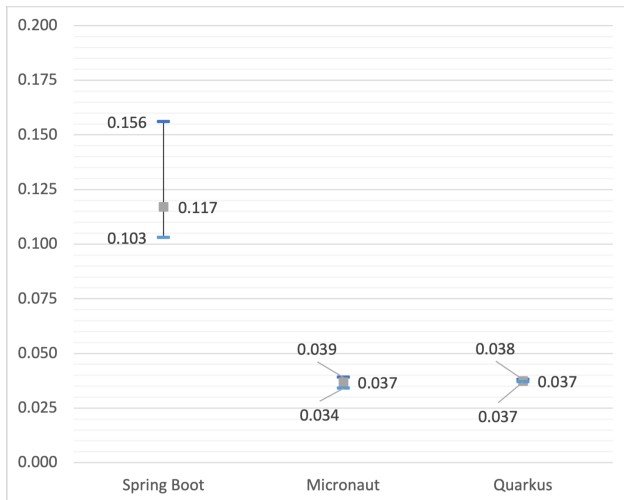

(**a**) JAR executable.    (**b**) Native image.

**Figure 4.** Application startup time [s].

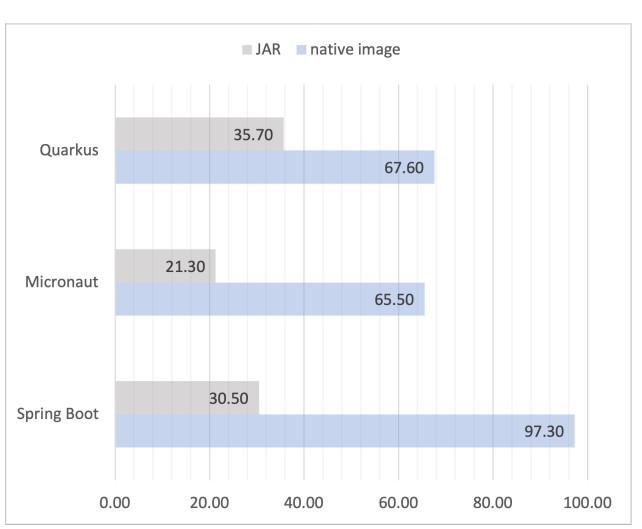

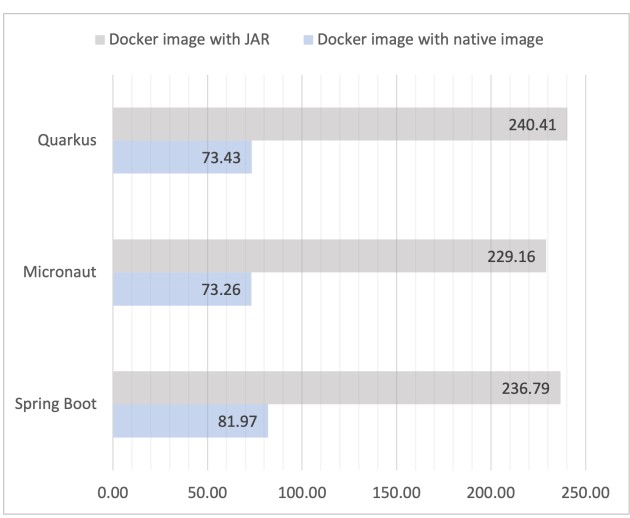

(**a**) Executable files.    (**b**) Docker images.

**Figure 5.** Size of Docker images and executables [MB].

*4.4. Stress Tests Results*

The results of the executed tests scenarios that were categorized as stress tests are presented in Figures accordingly:

- SingleGreeting—Figure 6,
- GreetingSSE—Figure 7,
- CreateFetchDelete—Figure 8,
- MediumNumberSet—Figure 9.

It is worth mentioning that in the case of the CreateFetchDelete test the native image version of the Micronaut framework was excluded from the comparison due to an overwhelming number of failures compared to the scenario's successful executions (see Figure 8c). A similar situation has been observed in the case of MediumNumberSet for Spring Boot (native image) and Micronaut (both JAR and native image) (see Figure 9c). The same approach was used.

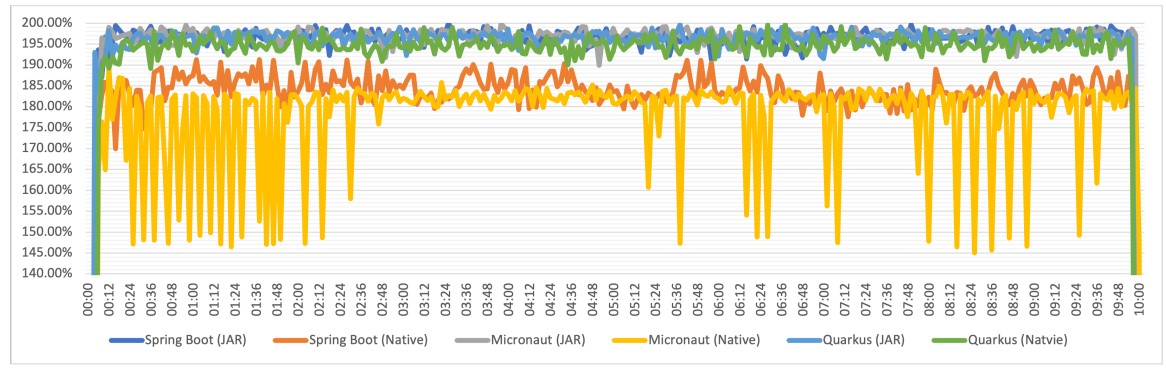

(**a**) Average CPU usage [%].

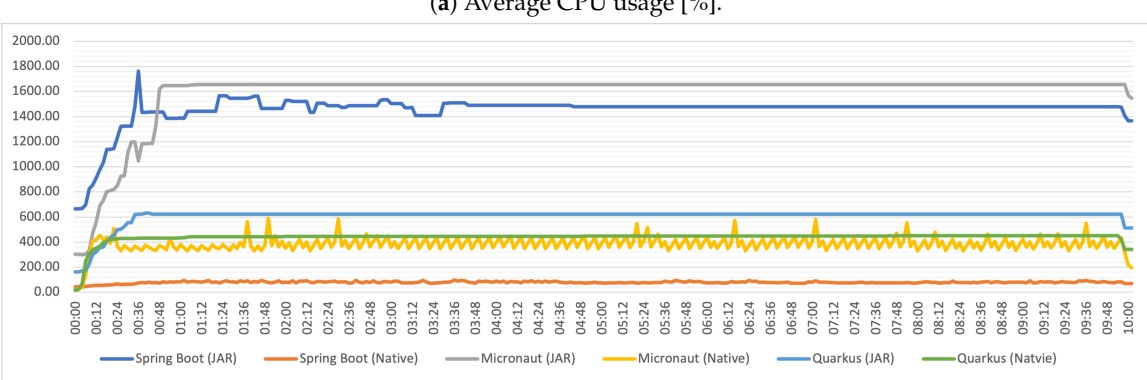

(**b**) Average RAM usage [MiB].

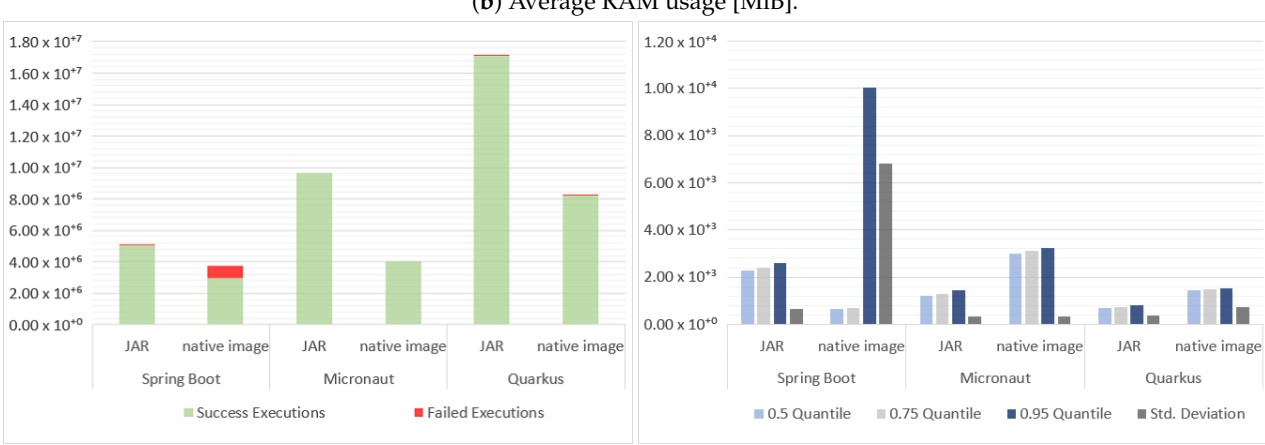

(**c**) Number of failed and success scenario executions.

(**d**) Quantile's and std. deviation of response time [ms].

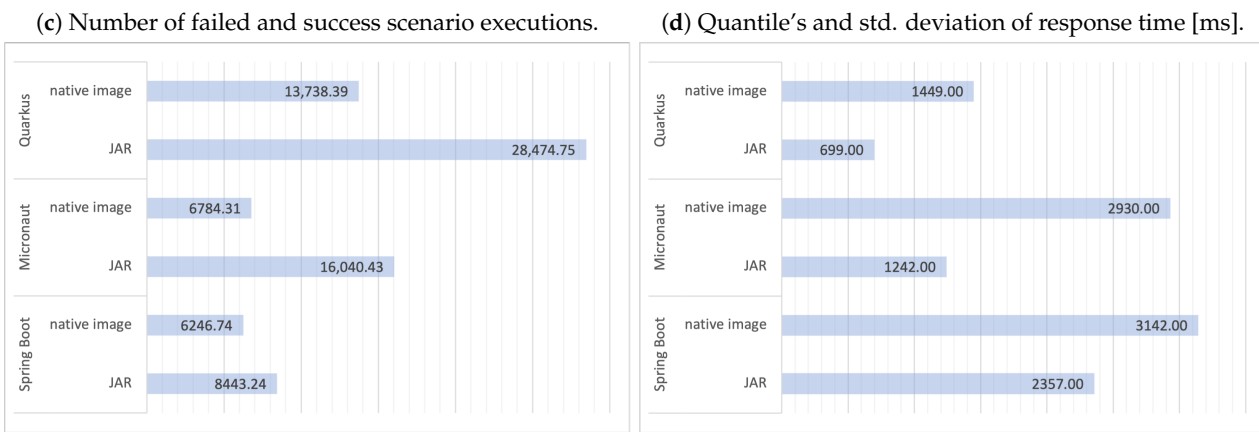

(**e**) Number of scenario executions [cnt/s].

(**f**) Average response time [ms].

**Figure 6.** SingleGreeting test results.

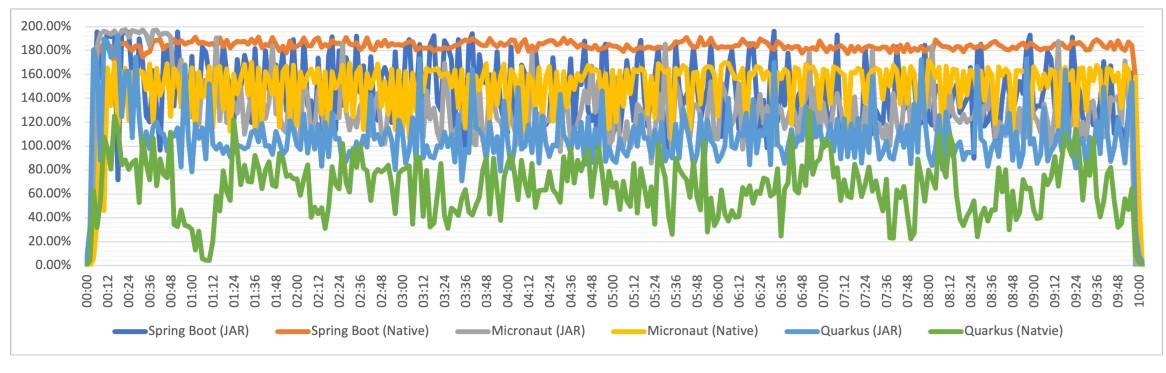

(**a**) Average CPU usage [%].

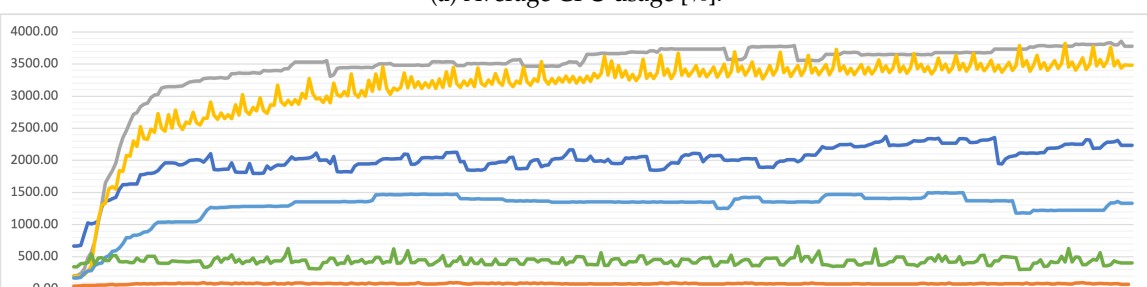

(**b**) Average RAM usage [MiB].

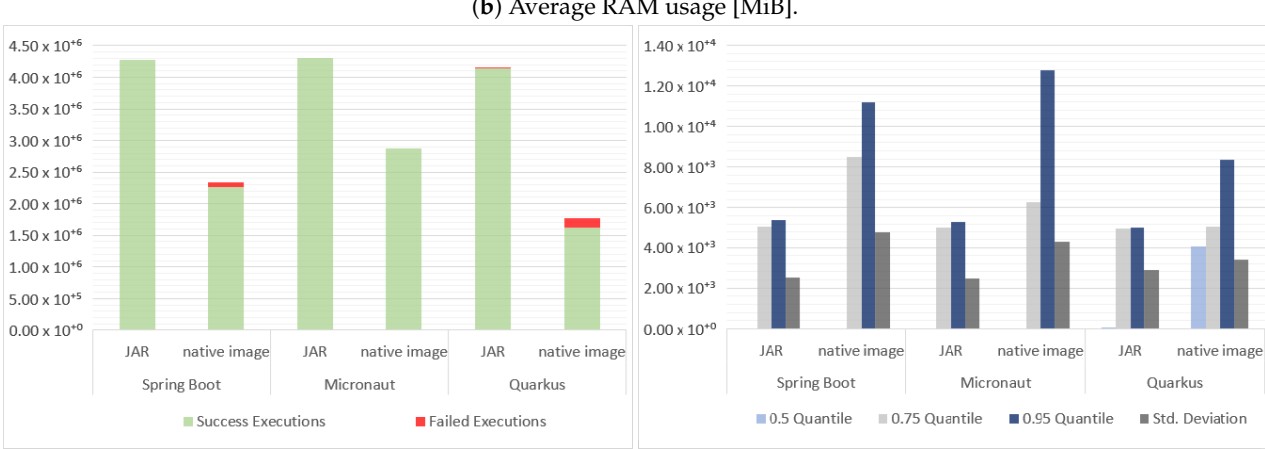

(**c**) Number of failed and success scenario executions.     (**d**) Quantile's and std. deviation of response time [ms].

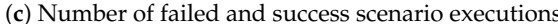
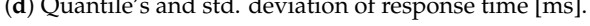

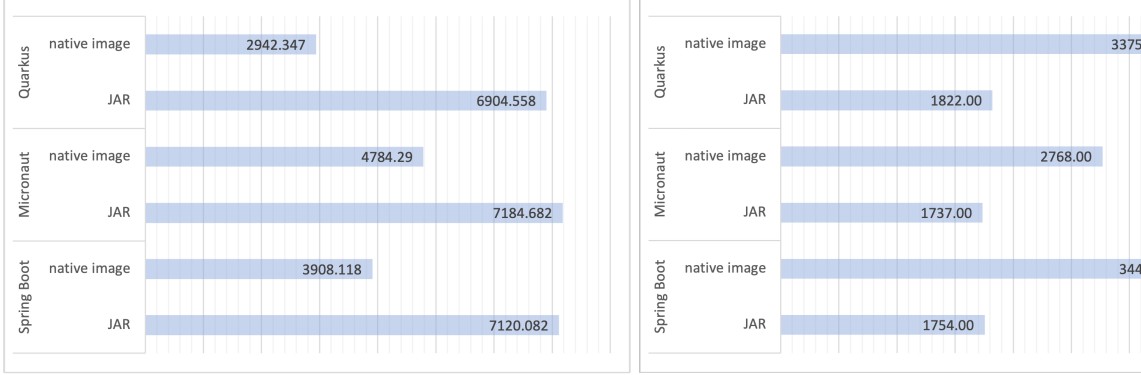

(**e**) Number of scenario executions [cnt/s].       (**f**) Average response time [ms].

**Figure 7.** GreetingSSE test results.

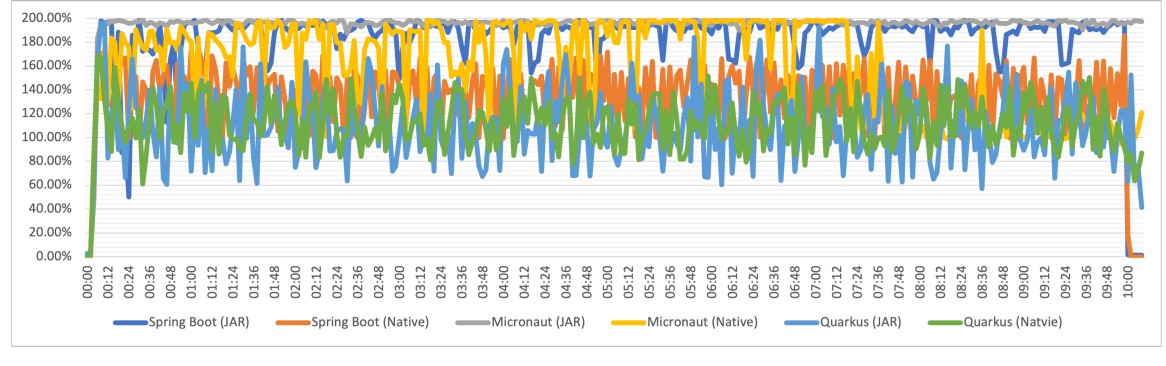

(**a**) Average CPU usage [%].

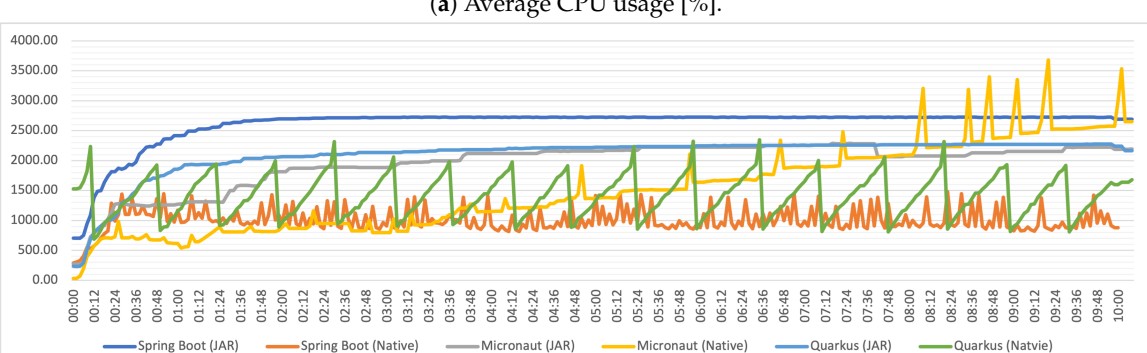

(**b**) Average RAM usage [MiB].

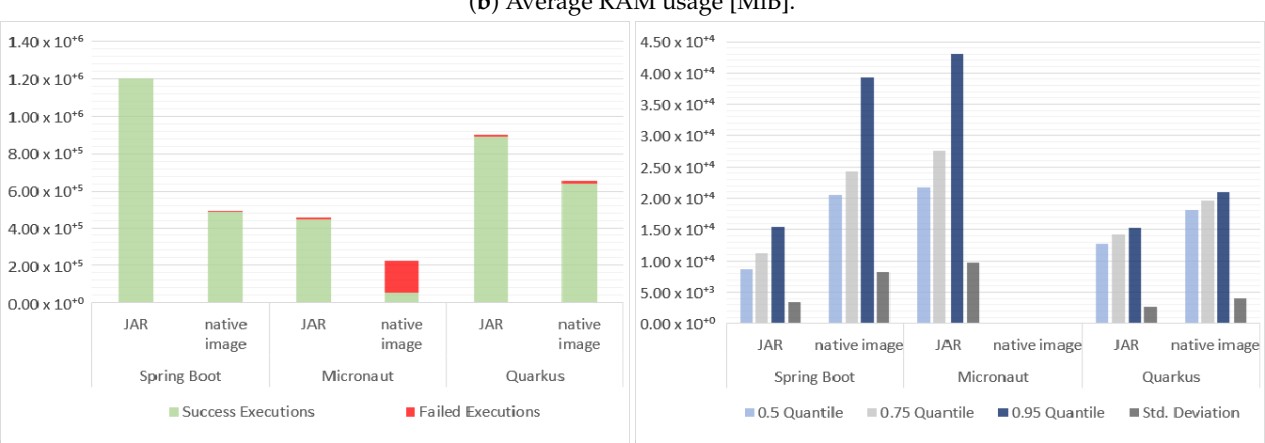

(**c**) Number of failed and success scenario executions.

(**d**) Quantile's and std. deviation of response time [ms].

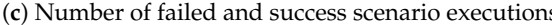

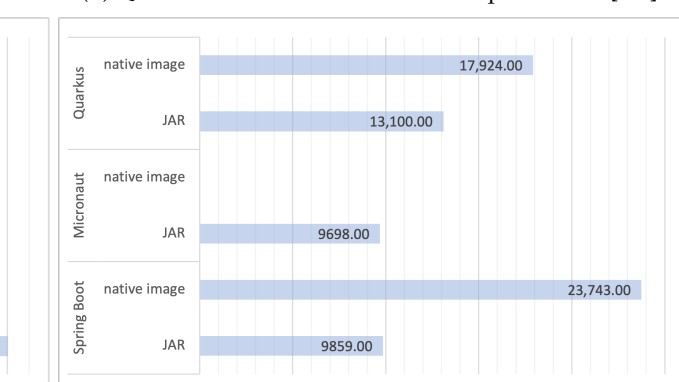

(**e**) Number of scenario executions [cnt/s].

(**f**) Average response time [ms].

**Figure 8.** CreateFetchDelete test results.

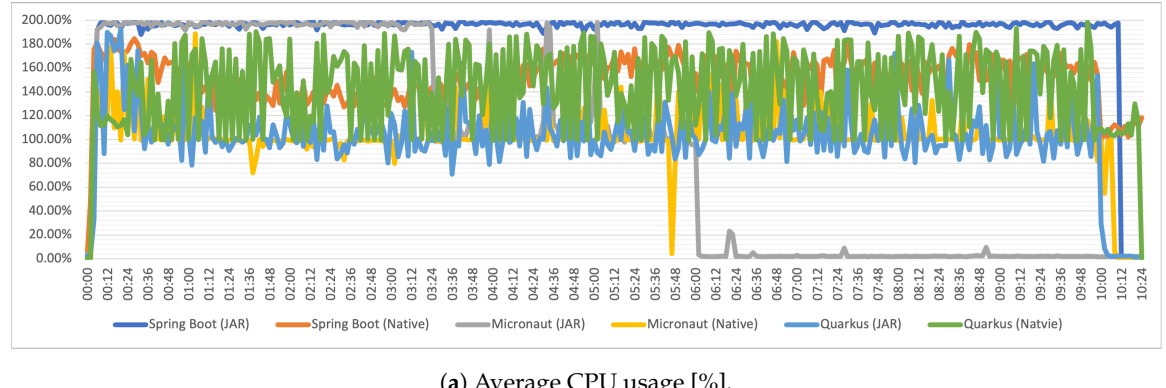

(**a**) Average CPU usage [%].

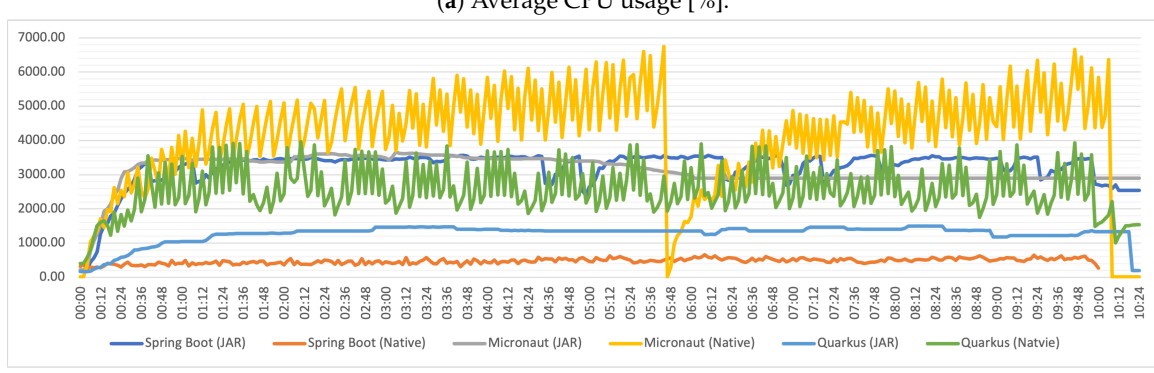

(**b**) Average RAM usage [MiB].

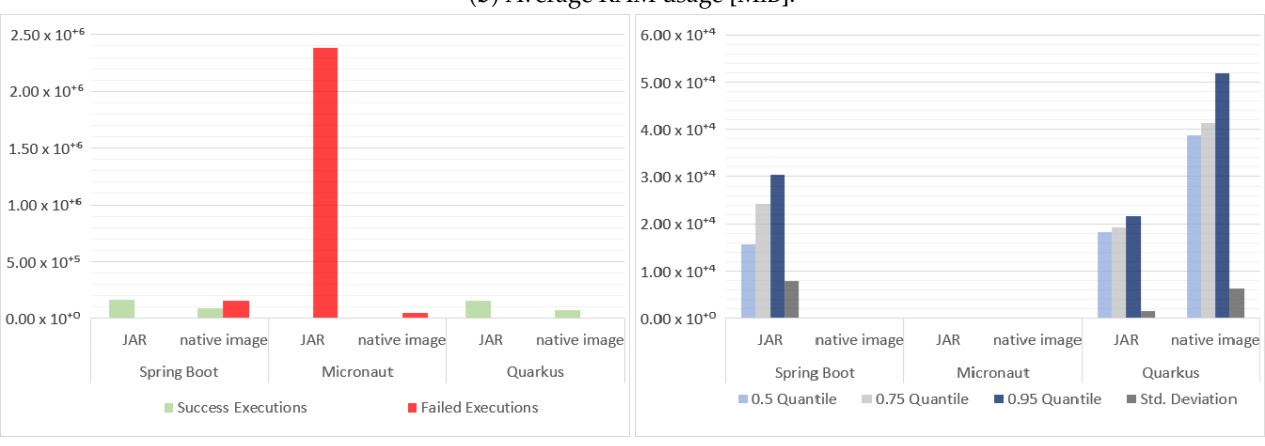

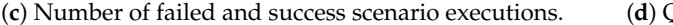

(**c**) Number of failed and success scenario executions.

(**d**) Quantile's and std. deviation of response time [ms].

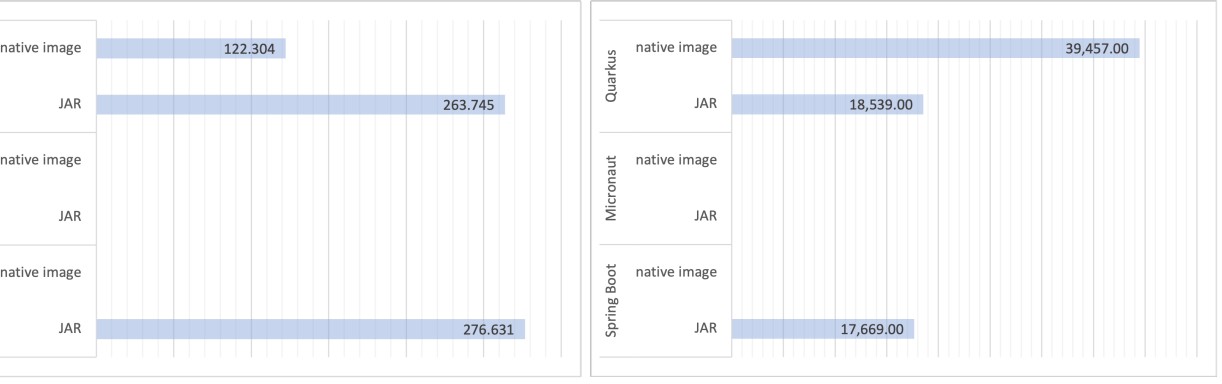

(**e**) Number of scenario executions [cnt/s].

(**f**) Average response time [ms].

**Figure 9.** MediumNumberSet test results.

### 4.5. Average Load Tests Results

The results of the average load tests are presented in Figures:

- SmallNumberSet—Figure 10,
- Notes—Figure 11,
- CreateFetchPatchDelete—Figure 12.

Charts for the failed/successful executions and execution count per time unit were left out as these values were constant for each test and did not provide relevant information from the point of view of the research.

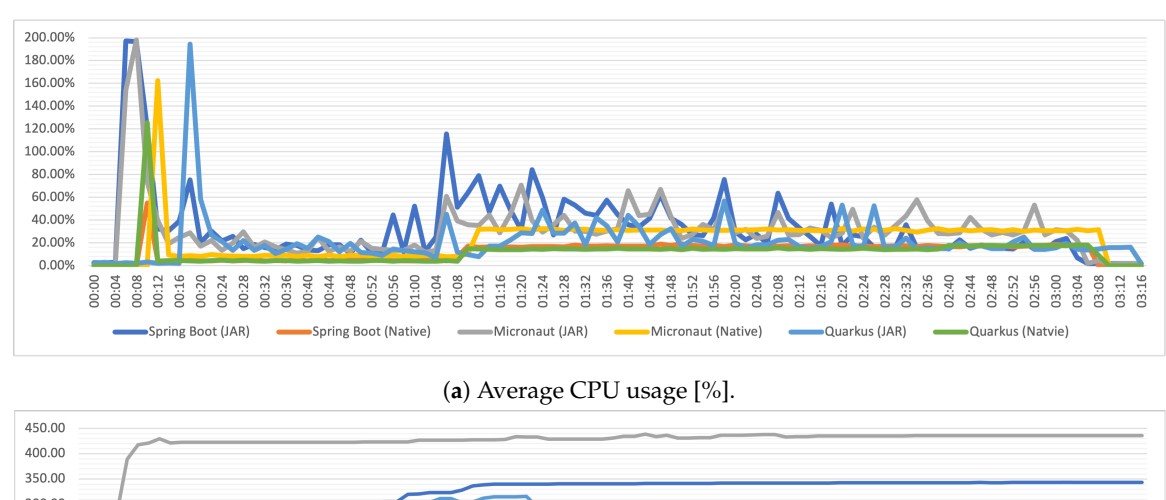

(**a**) Average CPU usage [%].

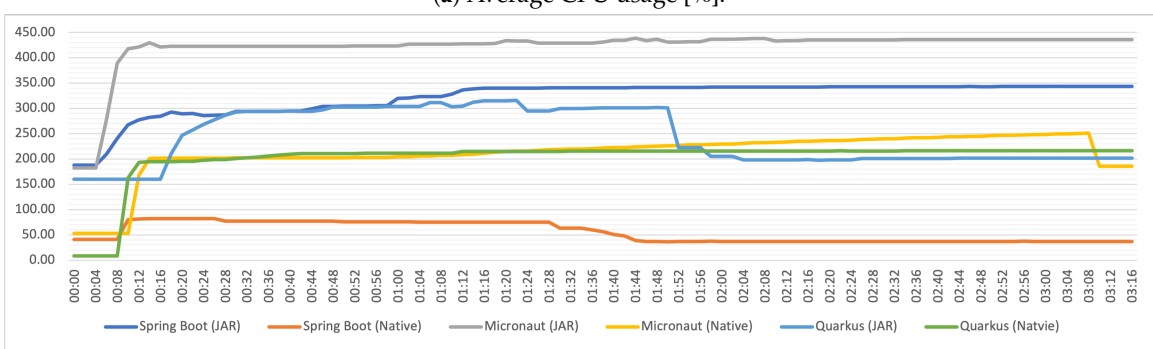

(**b**) Average RAM usage [MiB].

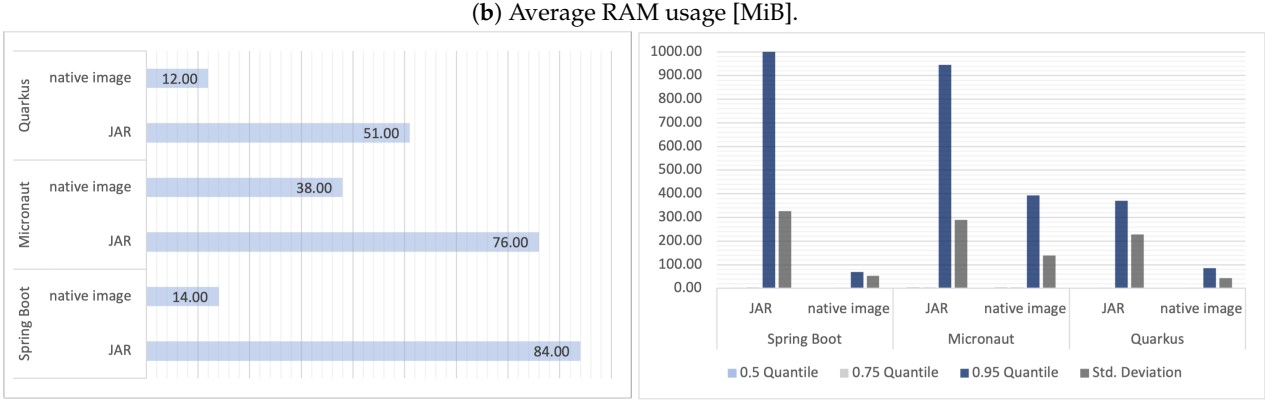

(**c**) Average response time [ms].      (**d**) Quantile's and std. deviation of response time [ms].

**Figure 10.** SmallNumberSet test results.

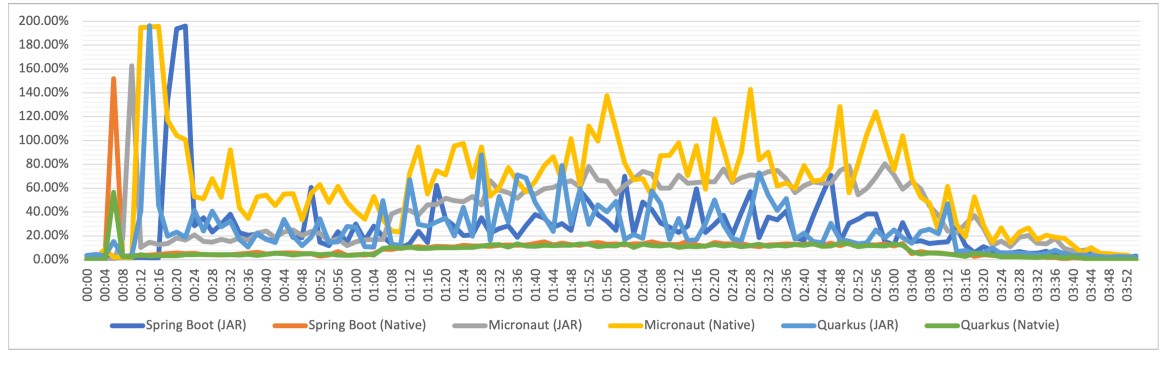

(**a**) Average CPU usage [%].

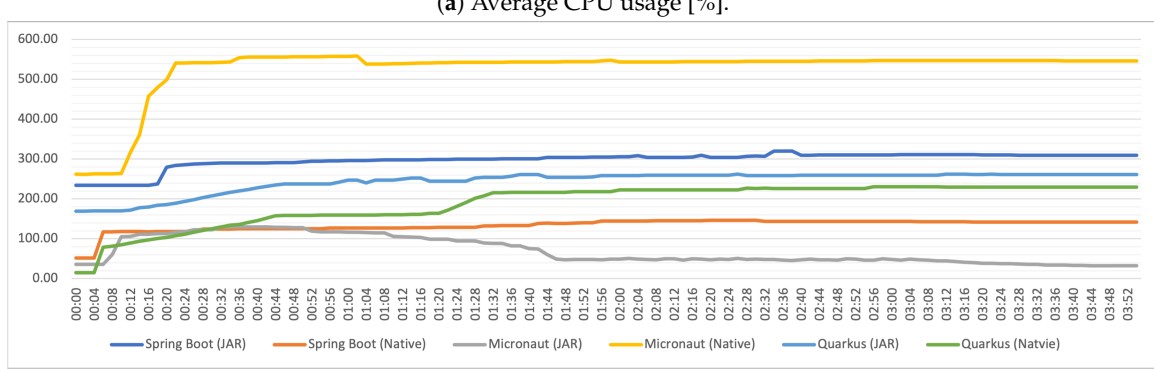

(**b**) Average RAM usage [MiB].

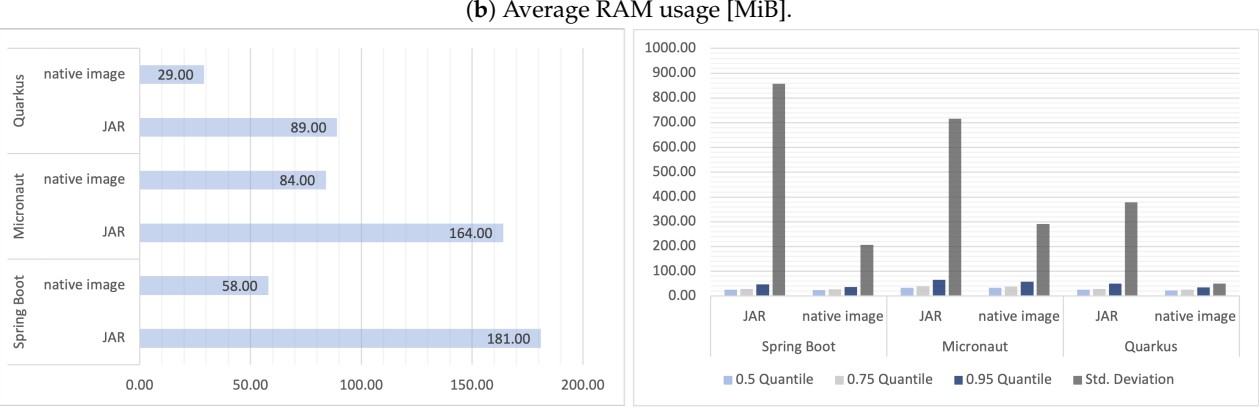

(**c**) Average response time [ms].

(**d**) Quantiles and std. deviation of response time [ms].

**Figure 11.** Notes test results.

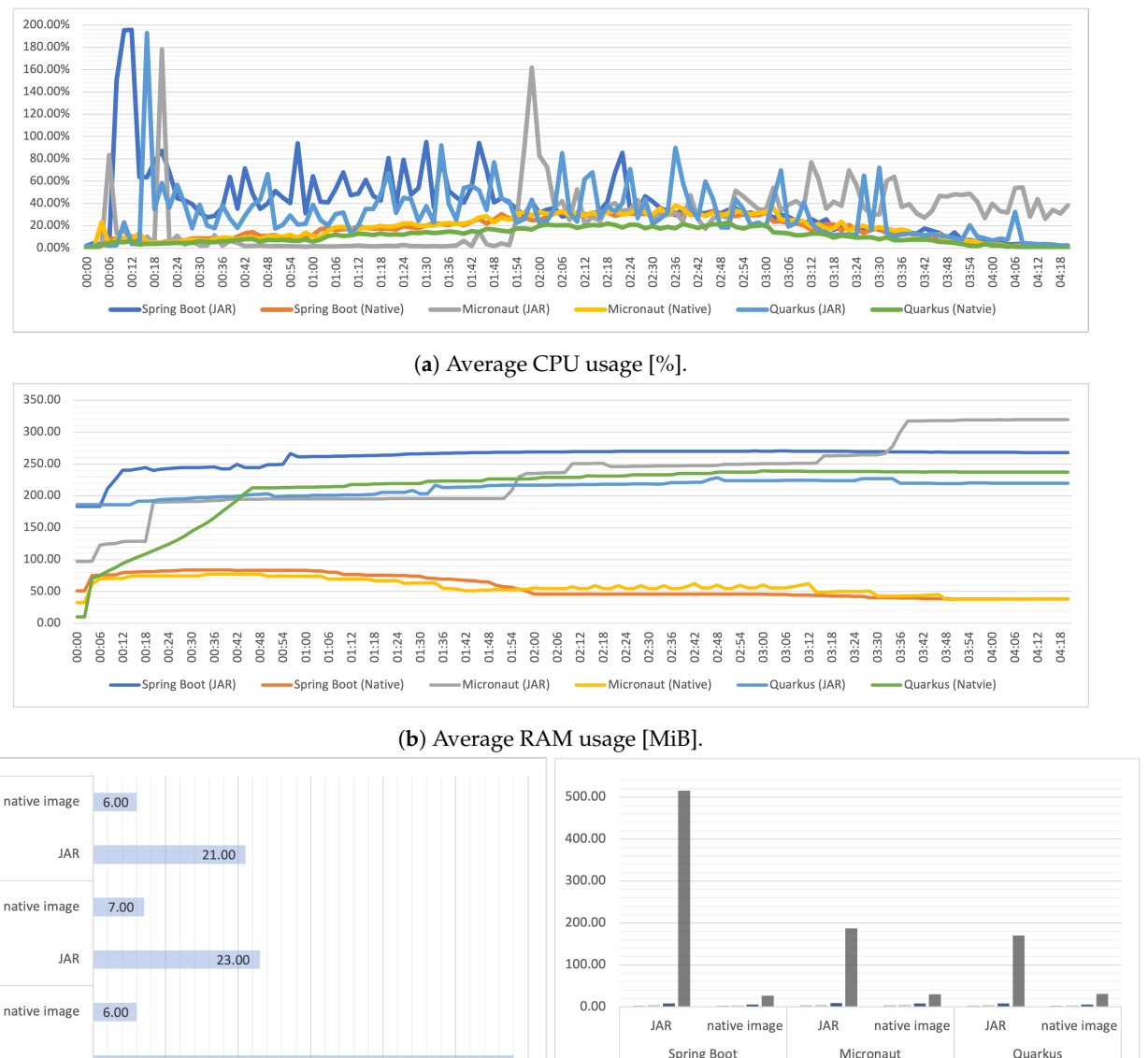

(**a**) Average CPU usage [%].

(**b**) Average RAM usage [MiB].

(**c**) Average response time [ms].

(**d**) Quantile's and std. deviation of response time [ms].

**Figure 12.** CreateFetchPatchDelete test results.

## 5. Discussion

In this study, the objective was to evaluate comparatively and critically the most promising JVM frameworks supporting the development of microservices. To this end, synthetic benchmarks were developed to reflect the most common uses of web services. Then performance tests of various workloads were carried out in a controlled execution environment using these benchmarks. The empirical results obtained allowed us to make some interesting observations.

Obtained results and insights learned can be analyzed and presented in many ways. A decision has been made to show the results in two contexts-—a general one that will focus on the main differences noticed between analyzed frameworks and in the context of the comparison between the usage of the traditional JAR executable files approach and the emerging new technique of the native image files still considering studied frameworks as well.

Starting from the factors that were noticed during the application preparation phase, we can see that Spring Boot leads the way in terms of documentation. The much younger competitors, for now, cannot stay ahead of their older rival regarding the accuracy, exten-

sivity and the number of examples available in the documentation (but on the Internet itself as well).

Analyzing the compilation time under test (Figure 3), we can see the different approaches taken in the Micronaut and Quarkus directly impacting the results and translating into the application start time. Spring Boot relies on the reflection and proxies in its IoC container implementation and uses it in the dependency injection process. Based on the Spring framework experience and awareness of the burden the usage of runtime-based techniques combined with reflection and proxies carries, the others tend to minimize their use in their IoC/DI solutions and leverage ahead-of-time compilation as much as possible. It is worth noting that Spring also starts to go towards that direction with the announced initial AOT support and optimization at the build time forthcoming with Spring 6.0. Independently from the executable format, we can see the benefits of that approach which, however, comes at a price. The compilation time for JAR files is noticeably longer for the Micronaut and Quarkus as they shift most of the operations that (in the case of Spring Boot) are done in runtime to the compilation time. Nevertheless, this gives the significant benefit of a shorter startup time (Figure 4a) that can be crucial from the point of view of autoscaling. We can see a bit of a different effect for the compilation time in the native image (Figure 3b). Higher reliance on the reflection causes an increase in it, as it must be registered upfront and probably requires additional operations during compilation. Yet, as in the case of the JARs, the achieved startup times by Spring Boot are worse than the competition (Quarkus and Micronaut) (Figure 4b). Moreover, regarding the compilation time for the native image, we can see that both Quarkus and Micronaut achieved almost the same and the best result. When considering the JAR executable, Micronaut is the leader. The usage of the custom fast-jar approach in the Quarkus, which for some time has been the default approach in the framework, still is a bit behind the fastest competitor. As part of the observations made, it is worth noticing that Micronaut, by default, uses lazy initialization. It results in a much longer time to handle the first request (around 1–3 s) as the underlying components, such as services, etc., are only created. Possibly, these could impact the visible dominance of the Micronaut in terms of the startup time for the JAR executable.

Moving on to the topic of the executable file and the image size (Figure 5), regarding the JAR executable, we can see that Micronaut is the leader, with the result lower by over 10 MB from the most extensive Quarkus JAR file. When considering the native image, we can see that the executable size is at least twice as large and even three times as large in the Spring Boot case, which is the worst here and stands out from the others, for which the differences are minor (in favour of Micronaut). A larger executable size for the native image does not translate to a larger image size. In the case of the native image, we do not need to include Java virtual machine or Java Development Kit. It allows us to generate images that are approximately three times smaller. We can see here clear advantages both from the point of view of autoscaling—we have fewer megabytes to download so that we can do it faster, lowering the time needed to start new application replica, and from the end of the IoT device, which generally has more limited memory resources than for example servers hosting containers.

The stress tests performed allow us to draw many interesting conclusions. At the outset, it can be stated that in the case of stress tests, despite the competitive results of the average response time, the Micronaut framework was the worst due to the dominant problems in the MediumNumberSet (Figure 9) or CreateFetchDelete tests (Figure 8)—it did not cope with the established extreme load. It is also worth noting that in the case of such an excessive load in terms of the number of executions of the scenario and the average response time for each of the analyzed frameworks, the version in the form of a JAR file fared better than the corresponding native image. However, in some cases, this was associated with visibly greater use of the server's resources on which the applications were run. The stress test results do not indicate the dominant leader among the solutions tested. Depending on the test case, it was Quarkus; otherwise, Spring Boot was slightly better.

Subsequent average load tests showed an interesting result. In their case, the native image application performed better for each framework. In connection with the lower use of resources, it can be concluded that by using a well-functioning automatic scaling mechanism that would prevent the emergence of extreme situations, which were the subject of the previous tests, we could use this executable form of the application to obtain a more efficient system. The mean response time results obtained in these tests show much more pronounced differences between the templates in the case of JAR files. In this respect, Quarkus fared best. It is worth noting that in some cases, the Micronaut template obtained similar results.

The authors of this article tried to make the research as reliable as possible. The conducted research as well as the conclusions drawn on their basis do not exhaust the discussed issue. They can be continued in many directions. It would be worth considering the aspect of which of the executable forms of applications and frameworks is best suited for use in IoT devices as a client application, and whether, thanks to the low use of resources and relatively low price, these devices could be an alternative to building a cluster hosting a server application. An important direction of research development could also be the extension of the comparison to applications made in the same frameworks but using a blocking programming model and the Spring Boot framework using a servlet-based engine, for example, Tomcat.

## 6. Conclusions

In conclusion, it is worth noting that the technologies studied here support efficient software development in a microservices architecture, which in turn fosters horizontal scaling of system areas that require it. Since horizontal scaling offers almost unlimited possibilities to meet the system load, each of the technologies tested is a major step toward the development of global and high-availability IT systems. However, when implementing scaling and keeping in mind the cost-effectiveness of the implemented solution, it is worth considering the nature of the load variability over time and the overall computing power requirements of a given system. In the case of high load variability, the ability of the microservice scaling mechanism to keep up with this load will be important; thus, reducing the provisioning time and cold start delays of a new microservice instance. On the other hand, for systems with low volatility but high demand for computing power, computational efficiency, measured by the number of transactions processed per unit of time, will be of key importance. Then, one should choose a technology that makes the most optimal use of computing resources, which will ensure overall maximum performance.

In terms of performance, it can be said that the Quarkus template and its older colleague, Spring Boot performed very well. Micronaut was also able to achieve competitive results, but the problems seen in the stress tests caused it to lag behind the competition.

In terms of the executable form of the application, an alternative in the form of a native image may, in many cases, turn out to be competitive with classic JAR files. The most important advantage turned out to be the ability to significantly reduce Docker images, which reduces a microservice provisioning time. In addition, the application startup time has been shortened. However, these advantages are paid for by the longer compilation time or the requirement to register the use of reflection before compilation. Native images are not yet an ideal solution and will take some time to reach full production-grade maturity and reliability.

**Author Contributions:** Conceptualization, Ł.W. and Ł.L.; methodology, Ł.W. and Ł.L.; software, Ł.L.; validation, Ł.W. and Ł.L.; writing—original draft preparation, Ł.W. and Ł.L., and A.M.K.; writing—review and editing, Ł.W. and Ł.L., and A.M.K.; visualization, Ł.L. and A.M.K. All authors have read and agreed to the published version of the manuscript.

**Funding:** This work was supported by University of Silesia in Katowice (Institute of Culture Studies) and partially by Statutory Research funds of Department of Applied Informatics, Silesian University of Technology, Gliwice, Poland (BK/2023).

**Institutional Review Board Statement:** Not applicable.

**Informed Consent Statement:** Not applicable.

**Data Availability Statement:** Not applicable.

**Conflicts of Interest:** The authors declare no conflicts of interest.

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
