# Peer review of "A Comparative Assessment of JVM Frameworks to Develop Microservices"

_applsci, doi:10.3390/app13031343_

Round 1

Reviewer 1 Report

The authors present a comparison of Quarkus, Micronaut, and Spring Boot. The work addresses a very interesting and timely topic. The paper is well written (just some typos throughout the paper) and well organized. Results are robust.

Please, I strongly recommend the authors include the code that has been used for the different tests.

Reviewer 2 Report

The paper is globally well written..

However, the paper suffer from several problems as follows:

1-In the Introduction section, it is suggested that the authors address the following issues: 1) motivation, 2) goal and requirements of the solution you are presenting (start the sentence with "The goal of this work is..."), 3) why other solutions do not solve the problem (just a paragraph), 4) your solution, and 5) organization of the paper. 

So far, only the Motivation seems to be in this section.

2-Some citations are also lacking in Section 1.

3 - Still in Section 1, the following sentence "This is particularly important not only for deployed in large data centers server-based systems which provide their services to other clients including IoT devices but also for the need to increase the computing power of systems run at the edge (so-called edge computing), where individual computing nodes, often supporting IoT devices or being IoT devices themselves, have limited capacity." is clearly very long !

4- Section 2 should be in fact something similar to Related Work. However, there are some issues missing, in particular, say why you do not compare your study with others ! Is it better ? Such other work does not exist ? Say it explicitly !

5-In Section 3.2, why did you chose only three and why these three ? This is not clear at all !

6-In Section 3.2, you say that "The number of respondents to that surveys...". How many are we talking about ? Say it between parenthesis at least.

7-In Section 3.3, you say that "It can be seen that most comparative analysis that concerns similar topics use a set of performance tests to evaluate and compare analysed solutions". These should have been mentioned in the Related Work section (that you do not have).

8-When referring to tables (e.g., "...presented in tables 1 and 2." in page 7), you should have instead Table!

9-In Section 3.4, why 10 minutes ? Also, the other numbers you use (e.g., "250 concurrent users", "500 users spread out", etc...) ?

10-In Section 3.5, why Google Cloud? Others could have been used ! 

11-Also in Section 3.5, what kind of VMs ? Linux? Windows ? MacOS?

12-In some figures you have "JAR executable" and "Native image". Make clear the differences.

14-In Section 4.1 and Section 4,2, how many are "several" ?

15-I guess you want to say "measurements" and not "measures" in page 9.

16-In Figure 4b, there is a typo !

17-Section 5 should present the conclusions of the paper and not a discussion of the results obtained ! It should also be mentioned what table you are referring to with the results being considered.

18-The final paragraph is mostly a section on Future Work.
